# Prevalence of and Socio-Demographic Factors of Malnutrition Among Vietnamese Children and Adolescents: A Cross-Sectional Study

**DOI:** 10.3390/healthcare13060612

**Published:** 2025-03-12

**Authors:** Ngan Thi Duc Hoang, Nghien Thi Thao Hoang, Duong Thanh Tran, Hoa Anh Le, Tuyen Danh Le, Ewa A. Szymlek-Gay, Hiep N. Le, Hiep Thi Le, Du Thi Doan Dang, Hai Phung

**Affiliations:** 1National Institute of Nutrition, 48B Tang Bat Ho, Pham Dinh Ho, Hai Ba Trung, Hanoi 100000, Vietnam; tranthanhduong.ninvn@gmail.com (D.T.T.); leanhhoa.ninvn@gmail.com (H.A.L.); ledanhtuyen@gmail.com (T.D.L.); lethihiep.ninvn@gmail.com (H.T.L.); dangthidoandu.ninvn@mgail.com (D.T.D.D.); 2School of Medicine and Dentistry, Griffith University, Gold Coast, QLD 4215, Australia; ngochiep.le@griffithuni.edu.au (H.N.L.); hai.n.phung@griffith.edu.au (H.P.); 3Department of Clinical Nutrition and Family Medicine, Faculty of Medicine, VNU University of Medicine and Pharmacy, 144 Xuan Thuy Street, Cau Giay District, Hanoi 100000, Vietnam; nghienhtt.ump@vnu.edu.vn; 4Vietnam Dietetics Association, 48B Tang Bat Ho, Pham Dinh Ho, Hai Ba Trung, Hanoi 100000, Vietnam; 5Institute for Physical Activity and Nutrition (IPAN), School of Exercise and Nutrition Sciences, Deakin University, Melbourne Burwood Campus, 221 Burwood Highway, Burwood, VIC 3125, Australia; ewa.szymlekgay@deakin.edu.au

**Keywords:** Vietnam, nutritional status, nutritional policy, anthropometry, adolescent

## Abstract

**Background/Objectives:** Limited data exist on the nutritional status of Vietnamese children aged 5 years and older. This study aimed to (1) determine the nutritional status and (2) assess the associations between malnutrition and socio-demographic factors among children and adolescents aged 5–19 years old in selected provinces in Vietnam. **Methods**: A cross-sectional study was conducted on 3055 children aged 5–19 years old. Children’s weight and height were collected to calculate BMI for Age Z-score and Height for Age Z-score to determine the prevalence of overweight/obesity, stunting, thinness/underweight, stunted-overweight, stunted-underweight, at least one type of undernutrition, and at least one type of malnutrition. **Results**: The prevalence of stunting and thinness/underweight was around 10%, overweight and obesity was 14.5%, while the prevalence of malnutrition was 36.5%, with 19.7% of children experiencing at least one form of undernutrition. Significant associations were found between malnutrition and socio-demographic factors such as age, ethnicity, and the number of household possessions. Stunting was more prevalent among older children and those from lower-income households, whereas overweight and obesity were more common in wealthier families. **Conclusions**: These novel findings highlight the need for targeted interventions addressing both undernutrition and obesity in Vietnam’s diverse demographic groups.

## 1. Introduction

Childhood malnutrition, encompassing both undernutrition (underweight, stunting, and wasting) and overnutrition (overweight and obesity), remains a significant global challenge, particularly in low- and middle-income countries [1]. Recent estimates highlight the coexistence of undernutrition [2] and overnutrition [3] among preschool children, a trend that continues into adolescence. Despite ongoing national policies, Vietnamese children continue to experience a significant dual burden of malnutrition, with both undernutrition and obesity on the rise. Notably, the prevalence of stunting among children under five remains a significant public health concern, reaching 19.6% in 2020 [4,5]. In addition, the rates of overweight and obesity among Vietnamese school children are alarming. For example, a 2016 study involving 2334 primary school children from rural areas in Vietnam found that 22.1% were overweight/obese, and 31.0% had abdominal obesity [6]. While previous research has focused on younger children, there are limited data on malnutrition among Vietnamese adolescents (5–19 years old). This study fills that gap by analyzing prevalence and socio-demographic associations across three provinces.

Undernutrition is a leading cause of child mortality [7,8] due to its impact on immune competence and susceptibility to infections such as pneumonia, diarrhea, and measles. Conversely, childhood obesity is a risk factor for adult obesity and is associated with chronic diseases like cardiovascular disease, type 2 diabetes, cancer, and other obesity-related disorders [9]. Meanwhile, parents’ demographic characteristics impact children’s eating habits in a positive way: parents with higher income and education were more likely to have children with healthier dietary behaviors [10]. Demographic elements are crucial for addressing child poverty from the nutrition perspective [11]. A good nutritional status and the absence of any type of malnutrition (under- and overnutrition) are important factors associated with the well-being of children [11,12]. Therefore, addressing this dual burden of malnutrition requires substantial efforts from the healthcare system.

Several policies and strategies have been implemented to address the nutritional challenges faced by children in Vietnam. The National Nutrition Strategy for 2021–2030, with a vision toward 2045, aims to improve Vietnamese people’s diet in terms of quantity and quality, reduce child malnutrition, and manage obesity/overweight [13]. This strategy emphasizes a multi-sectoral approach involving various stakeholders such as the Ministry of Health, Ministry of Agriculture and Rural Development, and international organizations like UNICEF and the World Bank. Vietnam has implemented national programs such as school meals and school milk programs to improve the nutritional status of school children. However, there is limited evidence of their effectiveness. Therefore, research focusing on older children (aged 5 years and older) is crucial to identify those at risk of malnutrition. This study aimed to (1) determine the nutritional status of children and adolescents aged 5–19 years old and (2) assess the associations between malnutrition and socio-demographic factors among children and adolescents aged 5–19 years old in selected provinces in Vietnam.

## 2. Materials and Methods

### 2.1. Study Design

A cross-sectional study was conducted between December 2020 and May 2021 on 3055 children aged 5–19 years old in three provinces in Vietnam.

### 2.2. Sample Size

The sample size was calculated using data from our previous study, where the prevalence of overweight/obesity was 22.1% [6]. With a 95% confidence interval, a 5% margin of error, a design effect of 3, and accounting for a 15% refusal rate, the desired sample size was 992 children per province, resulting in a total sample size of 2977 children across three provinces. At the end of the sampling stage, 3055 children agreed to take part in the study and signed the consent form.

### 2.3. Location Selection and Recruitment

Locations were randomly selected by a multistage random sampling strategy. Vietnam comprises three main geographical regions: Northern Vietnam, Central Vietnam, and Southern Vietnam. From the list of all provinces in each region, one province was randomly selected, and three districts were randomly selected from the list of districts in each province. These provinces were selected to capture diverse socioeconomic conditions and regional variations in malnutrition. In each district, the District Health Centre was invited to take part through written invitations followed by telephone calls and emails to explain the purpose of this study, confirm participation, and obtain written informed consent from the District Health Centre’s Director.

Subsequently, communes were conveniently selected. If a district or a commune declined to participate in this study, the next district or commune on the list was invited to ensure the desired sample size. By the end of the location selection stage, 18 communes had agreed to participate in the research.

The District Health Education and Training Department was contacted by the District Health Centre to obtain their agreement to organize the research in schools. Because the number of public high schools in each district was one to three schools, priority was given to selecting high schools and obtaining their consent first. Secondary schools, primary schools, and kindergartens, which were more numerous than high schools, were then selected until the desired sample size was reached in each district. If a District Health Centre or a school declined to participate, the next District Health Centre or school on the list was invited. By the end of the location section stage, 22 schools had agreed to participate in the research.

### 2.4. Participant Recruitment

Once a school agreed to participate in the research, all students at the school were invited to take part in the study. Teachers distributed the study information flyer and consent forms to the children to take home to their parents. Written consent from parents was then returned by the children to the teachers, who subsequently forwarded the consent forms to the research staff. Children were asked to provide their verbal assent. Children were excluded if they were younger than five or older than 19 years, if their parents or primary caregivers did not sign the consent form, if they had anthropometric abnormalities (e.g., severe scoliosis, which would not allow for the correct determination of height), if they had an intellectual impairment that would prevent them from understanding the aims of the study, or if they had chronic diseases that affected their ability to participate in anthropometric measurements and interviews.

### 2.5. Anthropometric Measures

Children were weighed in light clothing and without shoes with calibrated electronic body scales (SECA Robusta 813, SECA GmbH & Co. KG, Hamburg, Germany) and measured for height to the nearest 0.1 cm with a SECA stadiometer (SECA 222, SECA GmbH & Co. KG, Hamburg, Germany) according to standardized procedures [14].

BMI was calculated as weight in kg divided by height in meters squared. BMI for Age Z-score (BAZ) > +1 was used to define overweight and obesity, and BAZ < −2 was used to define thinness/underweight. Height for Age Z-score (HAZ) < −2 was used to define stunting [15]. Children were further classified as follows: stunted-overweight if they had HAZ < −2 and BAZ > +1, stunted-underweight if they had HAZ < −2 and BAZ < −2, experiencing at least one type of undernutrition if they had HAZ < −2 or BAZ < −2, or experiencing at least one type of malnutrition if they had HAZ < −2 or BAZ < −2 or BAZ > +1 [15].

### 2.6. Demographic and Socio-Demographic Data

Information on age, ethnicity, household size, the number of siblings, and household possessions was collected via a pretested questionnaire administered by the research staff to children and their parents. The possessions listed were as follows: computer, video compact disc player, digital video disc player, fridge, washing machine, air conditioner, motorbike, car, bicycle, and telephone.

### 2.7. Data Analysis

Data were collected from 3055 children; the response rate was 100%. Anthropometric data were analyzed using WHO Anthro Plus 2007 [16]. Data were cleaned and entered into EpiData (version 3.1) software.

Participants’ ages were categorized into quartiles (5 to <9 years old, 9 to <12 years old, 12 to <14 years old, and 14 to 19 years old). Ethnicity was categorized as Kinh or other groups. Household size was categorized as ≤3 people (2 parents and 1 child), 4 people (2 parents and 2 children), 5–6 people (2 parents, 2–3 children, and/or 1 other relative), and >6 people (parents, children, and other relatives). The number of siblings was categorized as ≤2 and >2 children. The household possessions variable was dichotomized at 6 based on the median split, ensuring an even distribution for analysis.

The prevalence of stunting, thinness/underweight, overweight and obesity, stunted-overweight, stunted-underweight, at least one type of undernutrition, and at least one type of malnutrition was estimated for the whole sample, and for the levels of the following factors: age, ethnicity, household size, number of siblings, and number of household possessions.

Generalized linear mixed models, with the school as a random effect, were used to estimate the association between binary outcomes (stunting, thinness/underweight, overweight and obesity, stunted-overweight, stunted-underweight, and at least one type of undernutrition or malnutrition) with socio-demographic variables. We report univariate associations, i.e., only one socio-demographic factor considered at a time, and adjusted associations estimated under models including all socio-demographic factors (i.e., age, ethnicity, household size, number of siblings, and number of household possessions).

Data completeness was verified before analysis, and missing values were managed using list-wise deletion to avoid bias.

All estimates are reported along with 95% confidence intervals (CI). Analyses were performed with Stata (version 14.0; StataCorp LP, College Station, TX, USA).

## 3. Results

### 3.1. Nutritional Status of Children Aged 5–19 Years Old in Some Provinces in Vietnam

The study included 3055 children, nearly 60% of whom were male. Approximately 8% of the children lived in households with fewer than two siblings, while the majority resided in families of four to six members (Table 1).

The prevalence of stunting was around 13%, while thinness/underweight affected approximately 9% of children. Importantly, around 40% of the children experienced at least one form of malnutrition, 20% were affected by undernutrition, and nearly 15% were classified as overweight/obese (Table 2).

### 3.2. Associations Between Malnutrition and Socio-Demographic Factors Among Children Aged 5–19 Years Old in Some Provinces in Vietnam

Results from univariate analyses revealed that stunting was significantly associated with ethnicity, age, number of siblings, and the number of household possessions. However, in the multivariate analysis, only age and the number of household possessions remained statistically significant (Table 3). The likelihood of being stunted increased with age but decreased with the number of household possessions.

Overweight and obesity were significantly associated with most of the defined factors in univariate analyses, except for household size. In the multivariate analysis, the number of siblings was not significantly associated (Table 4). Girls, children from other ethnic groups, and older children were less likely to be overweight or obese compared to boys, children from the Kinh ethnic group, and younger children. Conversely, children from families with more household possessions (indicative of better socioeconomic status) were at higher risk of being overweight or obese compared to those from families with lower socioeconomic status.

Other analyses for the associations of thinness/underweight, stunted-overweight, stunted-underweight, and at least one type of undernutrition or malnutrition with socio-demographic variables showed no significance.

While most findings were statistically significant, some associations such as ethnicity and overweight showed wide confidence intervals, suggesting variability across subgroups.

## 4. Discussion

### 4.1. Nutritional Status of Children and Adolescents Aged 5–19 Years Old

This study highlights the dual burden of malnutrition (DBM) among Vietnamese children aged 5–19 years old, with the prevalence of stunting and thinness/underweight at 9–13% and of overweight and obesity at 15%. These findings are similar to data for school children aged 12–15 years old in South Asian countries (approximately 10% of children are affected by thinness, stunting, or overweight) [17]. Other countries in Asia, especially those experiencing rapid urbanization and economic growth, such as India [18], China, Nepal, Indonesia, and Pakistan [19,20], are also grappling with this issue. However, the children in this study appear to be at higher risk of malnutrition compared to those in other South Asian countries, with approximately one-fifth facing at least one type of undernutrition and two-fifths facing at least one type of malnutrition. Meanwhile, a research study that comprised data from the Demographic and Health Survey in Bangladesh, India, Nepal, Pakistan, Myanmar, Timor, Maldives, and Cambodia (from 2007 to 2017) of 798,961 households found the pooled prevalence of at least one type of DBM (overweight or obese, wasted, underweight, stunted mothers and/or children) at 12.0% [21]. Although it is challenging to compare the data from the two studies due to differences in sample sizes and sampling methods, the findings from our study have raised an alert for public health nutrition in Vietnam, particularly for childhood overweight and obesity. Our research was conducted in three poor and remote provinces; however, the prevalence of overnutrition was still 15%, which is similar to the national level (19%) [4].

Vietnam has been on track with two indicators from Sustainable Development Goal 3 for child health and well-being [22], but the DBM has been clearly observed since the 2000s in children under five years of age [23]. This has been reported in different studies in children aged 0.5–11.9 years old in three geographic areas (2011) [24], as well as in primary school children in megacities such as Hai Phong City (2016) [6] and Ho Chi Minh City (2014–15) [25]. Results from the general nutrition survey in Vietnam in 2020 reported a prevalence of stunting and overweight/obesity among children aged 5–19 years old at 15% and 19%, respectively [4]. Collectively, the data demonstrate that the DBM tracks from preschool to school children in Vietnam and persists despite the nation’s efforts to improve nutritional status of its population.

This research demonstrates the presence of DBM among Vietnamese children aged 5–19 years old. Findings from our study indicate a widespread prevalence of DBM across the nation. A heightened risk of at least one form of malnutrition was reported despite ongoing national efforts to improve nutrition. The persistence of DBM highlights the urgent need for more targeted public health interventions to address both undernutrition and obesity in Vietnam’s evolving nutritional landscape.

### 4.2. Associations Between Malnutrition and Socio-Demographic Factors Among Children and Adolescents Aged 5–19 Years Old

The results from this research revealed statistically significant associations between malnutrition and children’s age, sex, ethnicity, and the number of household possessions (a proxy for household wealth).

The positive associations between stunting and children’s age, and negative associations between stunting and the number of household possessions, were similar to other research in Vietnam [6,26]. It is widely agreed that stunting in adolescents results from an accumulation of multiple factors such as inadequate nutrient intakes; early childhood stunting; poor maternal health during early pregnancy; chronic infections and diseases; poor water, sanitation, and hygiene; socioeconomic and cultural factors; and inappropriate physical activity levels [27,28,29]. Therefore, the risk of stunting increases with age in children. Meanwhile, children can be protected from stunting if their households have higher levels of economic conditions (as indicated by an increased number of household possessions). A higher economic status can ensure an adequate food supply and sustain proper care for children [30]. Moreover, household wealth also influences mothers’ cognitive health and decision-making process, including food choices for their children and the identification of potential risks in their current nutrition care practices [31]. These findings confirm the need to maintain or improve the current nutrition strategies in Vietnam for childhood stunting control, prioritizing early interventions (for children under five), and targeting children from lower socioeconomic backgrounds, particularly those in remote and poor areas or ethnic groups [13].

In contrast to stunting, overweight and obesity had positive relationships with the number of household possessions, which is consistent with other research in Vietnam [6,24,32,33] but opposite to the figures in developed countries [34]. The higher the socioeconomic status of the family, the more likely it is to ensure household food security and increase access to food; therefore, the higher the risk of children becoming overweight or obese. Furthermore, Vietnamese parents prefer “rounder” or “chubby” children [35], which may also increase the risk of children consuming excess energy. Indeed, these findings were further reflected in the associations between overweight/obesity and ethnicity. Minorities in Vietnam usually live at a lower socioeconomic status [36] compared to the Kinh group, which accounts for 85.3% of the Vietnamese population [37] and, thus, experiencing a lower prevalence of overweight and obesity. Whether childhood overweight/obesity and household socioeconomic status have a causal relationship cannot be determined, as this is a limitation of our current cross-sectional study. Therefore, further well-designed research studies are needed.

Age was found to be negatively associated with childhood overweight and obesity. Adolescents tend to be more physically active than preschoolers, as they focus more on their physique. They experience a growth spurt during puberty, which significantly changes their body composition with an increase in muscle mass, particularly in boys. Older children acquire more knowledge about balanced and age-specific nutrition obtained through public health programs and are also likely to be influenced by peer pressure and other social factors, particularly social media, leading to more conscious weight management behaviors (e.g., exercise and food choices). However, it is also noted that social media can have a negative influence on older children. It has been reported that social media may lead to eating behavior such as eating disorders (anorexia nervosa or bulimia nervosa) and poor dietary habits (increased fast food and sweetened beverages consumption) [38,39]. While early childhood obesity is common [40], as children get older, these factors, along with improved healthy eating practices, can reduce the risk of obesity.

Girls were less likely to be overweight and obese compared with boys. This result supports findings reported by other research in Vietnam [6] and many other countries, such as Canada [41], China [42], and Thailand [43]. The influences of sex (biology: body composition, sex hormones, leptin concentration, etc.) and gender (socio-culture: food choices, weight-related concerns, physique preferences, sedentary behaviors, and physical activity patterns, etc.) have been proposed to account for the differences in adiposity between girls and boys [44]. The Vietnamese have a preference for boys; therefore, girls may be less prioritized compared to boys [45]. Girls are more concerned with being thin, slim, good at house chores, and preparing to take care of the family compared to boys [44].

Nutrition education and behavior change, school-based interventions, fortification and supplementation programs, and the regulation of unhealthy foods are some possible approaches to addressing these issues [46,47]. However, multi-sectoral strategies are usually challenging due to economic disparities, urban-rural divides, and entrenched eating behaviors. Rapid urbanization and changing food systems further hinder efforts to address these issues simultaneously. Additionally, there is a concern about the effectiveness of coordination between sectors, including agriculture, health, and education, which is necessary for implementing effective nutrition strategies.

Vietnam has implemented several national nutrition programs and policies aimed at addressing childhood malnutrition. The National Nutrition Strategy for 2021–2030, with a vision to 2045, focuses on reducing stunting, wasting, and micronutrient deficiencies while also addressing the rising prevalence of overweight and obesity [13]. This strategy emphasizes the importance of multi-sectoral collaboration involving various stakeholders such as the health, education, agriculture, and social protection sectors. Notably, UNICEF supports Vietnam’s efforts by promoting nutrition-specific and nutrition-sensitive interventions, including supplementation, food fortification, and the integration of nutrition education into school curricula [48]. The strategy also highlights the need for robust monitoring and evaluation systems to track progress and ensure the effectiveness of interventions. Existing national programs, such as school meal initiatives, require rigorous evaluation to assess their effectiveness. Future interventions should prioritize early prevention strategies targeting both stunting and obesity, and multiple strategies can be used to address this critical public health concern.


**Strengths and Limitations**


This study has strengths and limitations worth noting. One of the strengths is its large sample size, which enhances the generalizability of the findings. The research had no cases of refusal. Additionally, the study’s focus on a wide age range (5–19 years old) provides a comprehensive overview of the nutritional challenges faced by children and adolescents in Vietnam.

However, there are several limitations to consider. As this was a cross-sectional study, causality cannot be inferred. Additionally, using household possessions as a wealth indicator may not fully capture socioeconomic disparities. Longitudinal studies are needed to better understand the temporal dynamics of these associations. Furthermore, the reliance on self-reported data for some variables may introduce bias, and future studies should aim to include more objective measures.

Despite the limitations, our study contributes novel valuable data on DBM among children aged 5–19 years old in remote areas of Vietnam. Future research should focus on longitudinal studies to explore the causal pathways between socio-demographic factors and malnutrition. Additionally, there is a need for intervention studies to evaluate the effectiveness of existing public health programs, such as the school meal and milk programs, in improving the nutritional status of schoolchildren. Research should also investigate the impact of culturally tailored interventions that address the unique dietary practices and preferences of different ethnic groups.

## 5. Conclusions

These findings underscore the urgent need for tailored public health interventions to combat both stunting and obesity, particularly among vulnerable socioeconomic groups in Vietnam. Insights from this study can guide policymakers and public health practitioners in developing effective strategies to improve the nutritional status and overall health of children in Vietnam.

## Figures and Tables

**Table 1 healthcare-13-00612-t001:** Characteristics of the participants.

		N	%	Mean	SE
Age (year)	5 to <9	741	24.3		
	9 to <12	889	29.1		
	12 to <14	763	25.0		
	14–19	662	21.6		
Sex	Male	1793	58.7		
	Female	1262	41.3		
Ethnicity	Kinh	2020	66.1		
	Others	1035	33.9		
Household size	≤3 people	213	8.3		
	4 people	1117	43.2		
	5–6 people	1036	40.1		
	>6 people	217	8.4		
Number of siblings of the child	≤2 children	1573	79.9		
	>2 children	396	20.1		
Number of household possessions	≤6	1766	57.8		
	>6	1289	42.2		
BAZ		3052		−0.34	0.1
HAZ		3053		−0.85	0.1

**Table 2 healthcare-13-00612-t002:** Prevalence of malnutrition among children aged 5–19 years old.

	N	n (%)	95% CI
Stunting *	3050	394 (12.8)	9.8; 16.4
Thinness/underweight **	3052	267 (8.7)	6.2; 12.2
Overweight and obesity ***	3052	460 (14.5)	10.6; 19.6
Stunted-overweight ^a^	3055	16 (0.5)	0.2; 0.9
Stunted-underweight ^b^	3055	72 (2.2)	1.5; 3.4
At least one type of undernutrition ^c^	3055	594 (19.7)	15.0; 25.4
At least one type of malnutrition ^d^	3055	1039 (36.5)	31.4; 41.9

* Height for Age Z-score (HAZ) < −2 [15]; ** BMI for Age Z-score (BAZ) < −2 [15]; *** BAZ > +1 [15]; ^a^ HAZ < −2 and BAZ > +1 [15]; ^b^ HAZ < −2 and BAZ < −2 [15]; ^c^ HAZ < −2 or BAZ < −2 [15]; and ^d^ HAZ < −2 or BAZ < −2 or BAZ > +1 [15].

**Table 3 healthcare-13-00612-t003:** Association of stunting and socio-demographic factors.

Variable		Raw Estimates		Univariate Analysis *			Multivariate Analysis **		
		Total	Stunting N (%)	OR (95%CI)	*p*-Value	Overall *p*-Value ***	OR (95%CI)	*p*-Value	Overall *p*-Value ***
Age (year)	5 to <9 (reference)	737	74 (10.0)	1		<0.001	1		0.007
	9 to <12	889	136 (15.3)	1.60 (1.16; 2.19)	0.004		1.80 (1.19; 2.71)	0.005	
	12 to <14	762	88 (11.6)	1.40 (0.98; 1.99)	0.063		1.30 (0.83; 2.05)	0.252	
	14–19	662	96 (14.5)	2.22 (1.55; 3.18)	<0.001		2.01 (1.25; 3.24)	0.004	
Sex	Male (reference)	1792	243 (13.6)	1			1		
	Female	1258	151 (12.0)	0.91 (0.72; 1.16)	0.458		0.95 (0.71; 1.27)	0.739	
Ethnicity	Kinh (reference)	2018	219 (10.9)	1			1		
	Others	1032	175 (17.0)	1.40 (1.01; 1.93)	0.044		1.31 (0.87; 1.99)	0.200	
Household size	≤3 people (reference)	213	31 (14.6)	1		0.064	1		0.339
	4 people	1115	125 (11.2)	0.71 (0.47; 1.11)	0.141		0.70 (0.42; 1.16)	0.163	
	5–6 people	1035	149 (14.4)	1.02 (0.67; 1.57)	0.919		0.91 (0.53; 1.58)	0.743	
	>6 people	217	27 (12.4)	0.83 (0.47; 1.48)	0.532		0.77 (0.37; 1.59)	0.474	
Number of siblings of the child	≤2 children (reference)	1570	177 (11.3)	1			1		
	>2 children	395	68 (17.2)	1.39 (1.00; 1.91)	0.047		1.11 (0.74; 1.67)	0.605	
Number of household possessions	≤6 (reference)	1766	281 (15.9)	1			1		
	>6	1289	113 (8.8)	0.64 (0.49; 0.83)	0.001		0.71 (0.50; 0.99)	0.048	

* Generalized linear mixed model including the covariate as a fixed effect and school as a random effect. ** Generalized linear mixed model including all socio-demographic factors as fixed effects and school as a random effect. *** Overall *p*-value for the association between the outcome and the socio-demographic factor.

**Table 4 healthcare-13-00612-t004:** Association of overweight and obesity and socio-demographic factors.

Variable		Raw Estimates		Univariate Analysis ^a^			Multivariate Analysis ^b^		
		Total	Overweight or Obesity N (%)	OR (95%CI)	*p*-Value	Overall *p*-Value ^c^	OR (95%CI)	*p*-Value	Overall *p*-Value ^c^
Age	5 to <9 (reference)	738	146 (19.8)	1		<0.001	1		<0.001
	9 to <12	889	169 (19.0)	1.00 (0.77; 1.31)	0.978		1.51 (1.02; 1.26)	0.042	
	12 to <14	763	99 (13.0)	0.68 (0.50; 0.92)	0.013		0.98 (0.63; 1.50)	0.909	
	14–19	662	46 (7.0)	0.26 (0.18; 0.38)	<0.001		0.46 (0.27; 0.80)	0.005	
Sex	Male (reference)	1790	334 (18.7)	1			1		
	Female	1262	126 (10.0)	0.51 (0.40; 0.64)	<0.001		0.50 (0.36; 0.68)	<0.001	
Ethnicity	Kinh (reference)	2017	376 (18.6)	1			1		
	Others	1035	84 (8.1)	0.53 (0.38; 0.75)	<0.001		0.56 (0.36; 0.86)	0.009	
Household size	≤3 people (reference)	213	38 (17.8)	1		0.278	1		0.804
	4 people	1117	153 (13.7)	0.76 (0.51; 1.15)	0.194		0.84 (0.51; 1.39)	0.509	
	5–6 people	1034	136 (13.2)	0.73 (0.49; 1.11)	0.142		0.83 (0.48; 1.44)	0.511	
	>6 people	217	37 (17.1)	1.00 (0.59; 1.68)	0.985		1.02 (0.51; 2.06)	0.952	
Number of siblings of the child	≤2 children (reference)	1572	213 (13.6)	1			1		
	>2 children	396	36 (9.1)	0.55 (0.37; 0.82)	0.003		0.64 (0.40; 1.01)	0.056	
Number of household possessions	≤6 (reference)	1764	216 (12.2)	1			1		
	>6	1288	244 (18.9)	1.61 (1.28; 2.03)	<0.001		1.51 (1.11; 2.05)	0.008	

^a^ Generalized linear mixed model including the covariate as a fixed effect and school as a random effect. ^b^ Generalized linear mixed model including all socio-demographic factors as fixed effects and school as a random effect. ^c^ Overall *p*-value for the association between the outcome and the socio-demographic factor.

## Data Availability

The data presented in this study are available upon request from the corresponding author due to private and legal regulations of the Ministry of Science and Technology (funder) and the National Institute of Nutrition–Ministry of Health, Vietnam (organization in charge of the research) regarding their research data.

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
