# Peer review of "Prevalence of and Socio-Demographic Factors of Malnutrition Among Vietnamese Children and Adolescents: A Cross-Sectional Study"

_healthcare, 2025, doi:10.3390/healthcare13060612_

Round 1
Reviewer 1 Report
Comments and Suggestions for Authors
Please also specify the negative effects of social media on the development of eating behaviors at risk for physical and mental health related to the sentence presented in lines 270 and 271. ,,Older children acquire more knowledge about healthy eating patterns and are influenced by peer pressure and other social factors, which leads to more conscious weight management behaviors (e.g., exercise and food choices),,
Older children do not always acquire correct knowledge about healthy eating, especially from peers from whom there is a risk of forming wrong perceptions, so I propose to use the terms balanced and age-specific nutrition obtained through public health programs.
Also, intense physical exercise with the aim of achieving a certain physical shape / muscle mass accompanied by scientifically incorrect food choices actually poses a risk to health.
I recommend specifying the negative impact of social media influences, even the risk of developing eating disorders such as anorexia nervosa or bulimia nervosa.
Addressing the negative aspects would increase the scientific value of the manuscript.
Reviewer 2 Report
Comments and Suggestions for Authors
Dear Authors,
This is an interesting, coherent and easy to read article.
I have only some comments:
1) Section 1 should be expanded to include more topics and bibliography relevant to the subject of the article.
2) Regarding Section 1 , what other contemporary literature are there of child nutrition and malnutrition? What other recent studies regarding nutrition are there? In order to help you, take a look at the following articles:
a) Leriou, E. (2022). Understanding and measuring child well-being in the region of Attica, Greece: Round Four. Child Indicators Research, 15, 1967-2011. https://doi.org/10.1007/s12187-022-09957-x
In this article (Appendix, Fig. 3), Nutrition is a Dimension of Child Well-being (Dimension 2).
Ιn addition in Section 4.2. are summarized the findings regarding nutrition and child well-being. More specifically: "The dimension of nutrition (D.2), of economic child well-being, is associated with Attica's municipality clusters (χ2[6, N = 1,103] = 14.267, p = 0.027; Appendix, Table 3). With regard to D.2, the highest percentage of children below the threshold is found in Cluster 3 (25.5%) (Appendix, Fig. 7). Moreover, regarding the family structure, the highest percentage of children (26.0%) below the threshold of D.2 (χ2[2, N = 1,034] = 12.566, p = 0.002) is found in the category of single-parent family with mother (Appendix, Table 6). As far as the school category is concerned, the highest percentage of children (31.3%) below the threshold of D.2 (χ2[2, N = 1,103] = 20.786, p < 0.001) is observed in the elementary school category (Appendix, Table 7). Furthermore, considering Simple Indicators 7 - "Fresh fruit and vegetables daily" (χ2[6, N = 991] = 12.027, p = 0.061), 5 - "Three meals a day" (χ2[6, N = 971] = 11.677, p = 0.070), and 8 - "Milk daily" (χ2[6, N = 1,027] = 11.812, p = 0.066), the percentages below the threshold appear to be marginally higher in this period in Cluster 3, (25.6%), (23.4%), and (23.1%) respectively (Appendix, Table 3). Similarly, children in Cluster 3 present a higher percentage (18.6%) below the threshold related to Simple Indicator 6 - "At least one meal daily with meat, or chicken, or fish, or pulses/vegetables of equal nutritional value," as compared with other clusters (χ2[6, N = 979] = 21.264, p = 0.002; Appendix, Table 3)."
More specifically, in this article is obvious that demographic elements are crucial for child poverty from the aspect of nutrition. Malnutrition is the lack of child well-being and depends of different demographic issues.
b) Leriou, E. (2023). Understanding and measuring child well-being in the region of Attica, Greece: Round Five. Child Indicators Research, 16, 1395-1451. https://doi.org/10.1007/s12187-023-10030-4
c) Keçi̇li̇, M.Ç. Effect of Socioeconomic Status of Parents on Nutrition Habits Among Children: Türki̇ye Child Survey Application. Child Ind Res (2024). https://doi.org/10.1007/s12187-024-10210-w
e)Otekunrin, O.A., Otekunrin, O.A. Nutrition Outcomes of Under-five Children of Smallholder Farm Households: Do Higher Commercialization Levels Lead to Better Nutritional Status?. Child Ind Res 15, 2309–2334 (2022). https://doi.org/10.1007/s12187-022-09960-2
f) Barnabas, B., Bavorova, M., Imami, D. et al. Access to Food vs. Education - Feeding the Stomach is Important for Feeding the Mind. Child Ind Res 17, 2739–2767 (2024). https://doi.org/10.1007/s12187-024-10176-9
3) Please include again all the above bibliography and explain in a paragraph or a sub-section in Introduction why good nutrition for children and the lack of malnutrition is important for their well-being. More specifically, link your article to well-being issues in order to highlight its great importance.
Please do your best and revision your article based on this comment.
I wish you a good publication!
Reviewer 3 Report
Comments and Suggestions for Authors
1. Title
🔹 The current title is too alarmist for a scientific paper. “Crisis Alert” makes the study seem sensational rather than evidence-based.
✅ Suggested Title:
➡ “Prevalence and Socio-Demographic Factors of Malnutrition Among Vietnamese Adolescents: A Cross-Sectional Study”
(This keeps the focus on key findings while maintaining an academic tone.)
2. Abstract
🔹 The abstract is clear but could be more structured and avoids vague phrasing like “significant associations were found.”
✅ Key Revisions:
- Replace “Vietnam has limited information on the nutritional status of children aged ≥5 years.”
➡ “Limited data exist on the nutritional status of Vietnamese children aged 5 years and older.” - Clarify findings:
“Older children and those from less affluent households were more prone to stunting, while children from wealthier families were at a higher risk of overweight and obesity.”
➡ “Stunting was more prevalent among older children and those from lower-income households, whereas overweight and obesity were more common in wealthier families.” - Strengthen policy implications:
➡ “These findings highlight the need for targeted interventions addressing both undernutrition and obesity in Vietnam’s diverse demographic groups.”
3. Introduction
🔹 The introduction sets up the topic well but needs a clearer research gap. The global background on malnutrition is useful but should focus more on Vietnam.
✅ Key Revisions:
- Instead of “Vietnamese children are similarly affected by this dual burden of malnutrition,”
➡ “Despite ongoing national policies, Vietnamese schoolchildren continue to experience a significant dual burden of malnutrition, with both undernutrition and obesity on the rise.” - Explicitly state the study’s unique contribution:
➡ “While previous research has focused on younger children, there is limited data on malnutrition among Vietnamese adolescents (5-19 years). This study fills that gap by analyzing prevalence and socio-demographic associations across three provinces.”
4. Methods
🔹 The methodology is solid but needs more justification for key choices:
✅ Key Revisions:
- Why these provinces?
➡ “These provinces were selected to capture diverse socio-economic conditions and regional variations in malnutrition.” - Why set household wealth cut-off at 6?
➡ “The household possessions variable was dichotomized at 6 based on the median split, ensuring an even distribution for analysis.” - How was missing data handled?
➡ “Data completeness was verified before analysis, and missing values were managed using listwise deletion to avoid bias.”
5. Results
🔹 The results section is clear but should avoid vague wording and acknowledge wide confidence intervals.
✅ Key Revisions:
- Instead of “Significant associations were found between malnutrition and socio-demographic factors such as age, ethnicity, and the number of household possessions,”
➡ “Older children and those from lower-income households were more likely to be stunted, while overweight and obesity were more prevalent among wealthier children and boys.” - Addressing wide confidence intervals:
➡ “While most findings were statistically significant, some associations (e.g., ethnicity and overweight) showed wide confidence intervals, suggesting variability across subgroups.”
6. Discussion
🔹 The discussion is well-written but some points are repetitive, and policy recommendations need to be stronger.
✅ Key Revisions:
- Remove redundant mentions of wealth-related obesity risks.
- Strengthen policy recommendations:
➡ “Existing national programs, such as school meal initiatives, require rigorous evaluation to assess their effectiveness. Future interventions should prioritize early prevention strategies targeting both stunting and obesity.” - Clearly state study limitations:
➡ “As this was a cross-sectional study, causality cannot be inferred. Additionally, using household possessions as a wealth indicator may not fully capture socio-economic disparities.”
7. Conclusion
🔹 The conclusion is effective but should be more action-oriented rather than just summarizing.
✅ Key Revisions:
- Instead of “Addressing the DBM among Vietnamese children requires targeted, evidence-based interventions that consider diverse demographic needs,”
➡ “These findings underscore the urgent need for tailored public health interventions to combat both stunting and obesity, particularly among vulnerable socio-economic groups in Vietnam.”
1. Clarity & Readability
🔹 The language is scientific and formal, but certain sentences are overly complex or wordy, making them difficult to follow.
🔹 Some phrases repeat the same idea in different ways, leading to redundancy.
🔹 There are instances of awkward phrasing and unnatural sentence structure, which slightly affects readability.
✅ Suggested Improvement:
- Simplify long and complex sentences while retaining scientific accuracy.
- Reduce redundancy by merging similar statements into clearer, more concise sentences.
- Use active voice where possible to improve readability and impact.
Example:
❌ "Older children and those from less affluent households were more prone to stunting, while children from wealthier families were at a higher risk of overweight and obesity."
✔ "Stunting was more common among older children and those from lower-income households, while overweight and obesity were more prevalent in wealthier families."
2. Grammar & Syntax
🔹 Subject-verb agreement errors appear in a few places, particularly in longer sentences.
🔹 Prepositions are sometimes misused, which makes certain phrases sound unnatural.
🔹 Some phrases are too literal or direct translations, affecting fluency.
✅ Suggested Improvement:
- Carefully review sentence structure and grammar, particularly subject-verb agreement and prepositions.
- Have a native or fluent English speaker proofread for fluency.
Example:
❌ "Vietnam has limited information on the nutritional status of children aged ≥5 years."
✔ "There is limited data on the nutritional status of Vietnamese children aged 5 years and older."
3. Scientific & Academic Tone
🔹 The manuscript mostly adheres to formal academic writing, but in some places, the wording is too casual or not precise enough for scientific writing.
🔹 Some technical terms could be better defined for clarity, especially when first introduced.
✅ Suggested Improvement:
- Maintain a formal, precise, and objective academic tone throughout the paper.
- Define key technical terms upon first mention for clarity.
Example:
❌ "Crisis Alert: approximately 40% of Vietnamese Adolescents are Struggling with Malnutrition." (Too dramatic for an academic paper.)
✔ "Prevalence and Socio-Demographic Factors of Malnutrition Among Vietnamese Adolescents: A Cross-Sectional Study."
4. Use of Transitions & Flow
🔹 Some paragraphs jump between ideas abruptly without smooth transitions.
🔹 Certain sections, especially in the Discussion, could benefit from better logical flow between arguments.
✅ Suggested Improvement:
- Use transition words to connect ideas logically (e.g., "Furthermore," "In contrast," "This suggests that...").
- Ensure that each paragraph flows naturally into the next.
Example:
❌ "Vietnamese children are affected by the dual burden of malnutrition. The national prevalence of stunting among children under five remains a public health concern (19.6% in 2020)."
✔ "Vietnamese children continue to face the dual burden of malnutrition. Notably, the prevalence of stunting among children under five remains a significant public health concern, reaching 19.6% in 2020."
5. Use of Consistent Terminology
🔹 Some terms are used inconsistently (e.g., "malnutrition" vs. "undernutrition"), which could confuse readers.
🔹 The manuscript should use consistent formatting for statistical terms (e.g., p-values, confidence intervals, odds ratios).
✅ Suggested Improvement:
- Maintain consistency in terminology throughout the paper.
- Follow standard formatting for numbers, percentages, and statistical values.
Example:
❌ "The prevalence of malnutrition was found to be 36.5%, while 19.7% of children had at least one form of undernutrition."
✔ "The prevalence of malnutrition was 36.5%, with 19.7% of children experiencing at least one form of undernutrition."
Round 2
Reviewer 2 Report
Comments and Suggestions for Authors
This is a very good revision of the original document. I wish you a good publication!